# Spike S2 Subunit: Possible Target for Detecting Novel SARS-CoV-2 Variants with Multiple Mutations

**DOI:** 10.3390/tropicalmed9020050

**Published:** 2024-02-15

**Authors:** Teerada Ponpinit, Yutthana Joyjinda, Weenassarin Ampoot, Siriporn Yomrat, Phatthamon Virojanapirom, Chanida Ruchisrisarod, Abhinbhen W. Saraya, Pasin Hemachudha, Thiravat Hemachudha

**Affiliations:** 1Thai Red Cross Emerging Infectious Diseases Health Science Centre, King Chulalongkorn Memorial Hospital, Bangkok 10330, Thailand; 6187788020@student.chula.ac.th (Y.J.); weenassarin.a@chulahospital.org (W.A.); siriporn.yom@chulahospital.org (S.Y.); phatthamon@gmail.com (P.V.); 6471062723@student.chula.ac.th (C.R.); abhinbhen.s@chula.ac.th (A.W.S.); pasin.h@chula.ac.th (P.H.); 2Division of Neurology, Department of Medicine, Faculty of Medicine, Chulalongkorn University, Bangkok 10330, Thailand

**Keywords:** SARS-CoV-2, spike S2 subunit, multiple mutations, detection

## Abstract

Novel SARS-CoV-2 variants have multiple mutations that may impact molecular diagnostics. The markedly conserved S2 subunit may be utilized to detect new variants. A comparison of 694 specimens (2019–2022) in Thailand using a commercial RT-PCR kit and the kit in combination with S2 primers and a probe was performed. Delayed amplification in ORF1ab was detected in one BA.4 omicron, whereas no amplification problem was encountered in the S2 target. There were no statistically significant differences in mean Ct value between the target genes (E, N, ORF1ab, and S2) and no significant differences in mean Ct value between the reagents. Furthermore, 230,821 nucleotide sequences submitted by 20 representative counties in each region (Jan–Oct 2022) have been checked for mutations in S2 primers and probe using PrimerChecker; there is a very low chance of encountering performance problems. The S2 primers and probe are still bound to the top five currently circulating variants in all countries and Thailand without mismatch recognition (Jun–Nov 2023). This study shows the possible benefits of detecting S2 in combination with simultaneously detecting three genes in a kit without affecting the Ct value of each target. The S2 subunit may be a promising target for the detection of SARS-CoV-2 variants with multiple mutations.

## 1. Introduction

Severe acute respiratory syndrome coronavirus 2 (SARS-CoV-2) is a novel zoonotic pandemic virus causing the coronavirus disease of 2019 (COVID-19), a newly emerging infectious disease that occurred in late December of 2019 [1]. Manifestations of the COVID-19 disease are variable, ranging from mild or moderate respiratory illness to severe illness with pneumonia, respiratory failure, and hyperinflammatory syndrome that can be fatal [2,3]. By the end of December 2023, COVID-19 had killed more than 6.9 million worldwide since its initial identification [4].

SARS-CoV-2 is an enveloped virus with a single-strand, positive-sense RNA whose genome has approximately 29.9 kb encoding replicase protein (nsp12 or RdRp), 16 non-structural (nsp1–16), and 4 structural proteins (S, E, M, and N) [5]. SARS-CoV-2 has mutated rapidly over time with many mutations in several genes governed by genetic changes in the exoribonuclease (ExoN) domain of nsp14 (nsp14-ExoN), altering the proofreading mechanism of the virus, accumulating mutations, and improving viral fitness for transmission. The prevalence of the infection, overdispersion, and prolonged infection may contribute to host jumping and highly mutated variants. In addition, vaccination may exert selection pressure and influence viral evolution [6,7,8,9,10]. 

Genetic mutations are pervasive among most viruses, often resulting in benign or attenuated phenotypes. However, certain mutations, notably observed in SARS-CoV-2 variants of concern (VOCs), can confer increased virulence. These mutations may lead to increases in the severity of the disease and its ability to spread from human to human, compromise vaccine effectiveness, and potentially affect tests, especially the commercially available real-time RT-PCR kit-designed primers and probe based on the original SARS-CoV-2 even though there are circulating novel SARS-CoV-2 variants with multiple mutations. The impact of mutations on the performance of real-time RT-PCR tests depends on the virus’s nucleotide sequence, the test’s design, and the virus’s prevalence in the population [11]. Real-time RT-PCR targets multiple target genes (E, N, ORF1ab, and RdRp) of SARS-CoV-2 according to the Centers for Disease Control and Prevention (CDC) and WHO recommendation, as well as E and N genes for screening, followed by the confirmatory ORF1ab and RdRp [12]. However, the novel variants contain more mutations in several genes than previous SARS-CoV-2 variants, resulting in false-negative results or delayed amplification in diagnostic primers and probe commonly used and in some commercial kits [11,13,14].

The spike (S) protein is a glycoprotein on the viral surface. It specifically binds to host receptors and plays an important role in viral entry, expanding host range and cross-species infection [15]. It consists of an S1 subunit spanning (14–685), a globular head, and an S2 subunit spanning (686–1273), as well as a conserved stalk. The variation of the S1 subunit has been observed both across and within coronavirus genera. The N-terminal domain (NTD) (14–305) and receptor-binding domain (RBD) are located in this region spanning 319–541 residues. The S2 subunit plays a crucial role in cell-membrane fusion containing a fusion peptide (FP) (788–806), heptapeptide repeat sequence 1 (HR1) (912–984), heptapeptide repeat sequence 2 (HR2) (1163–1213), the transmembrane (TM) (1213–1237), and cytoplasm (CP) (1237–1273). This subunit is markedly conserved and has a low mutation rate [16,17] (Figure 1A). The conservation analysis of the S protein presents numerous mutations that frequently occur in NTD, RBD, and furin cleavage sites [10,18]. Meanwhile, the most conserved regions are HR1 and HR2 in the S2 subunit [19,20]. Mutations around the furin cleavage site, pre-FP, and the HR1 of S2 have been reported [17,21,22]. The Q954H and N969K mutations are positioned om HR1 and are often found in later omicron subvariants such as BA.1, BA.2, BA.5, BQ.1.1, XBB, EG.5.1, FL.4, GE.1, HK.3, HV.1, and JN.1, etc. [23,24]. In addition, the P1143L mutation in GE.1, JN.1, and BA.2.86, which is positioned between HR1 and HR2, was found [24]. Although some mutations have been found in this area, the chance of finding them is of a very low frequency due to the heat map showing that mutations in the S2 subunit were low, from 0.000431 to 0.001284 (Dec 2019–Jan 2022) [25]. Thus, the latter seems to be a promising target especially for conserved positions on the connector domain between HR1 and HR2 for detecting emerging SARS-CoV-2 variants.

The Fosun COVID-19 RT-PCR detection kit (Fosun, Shanghai, China) is a commercial real-time RT-PCR test that detects SARS-CoV-2 RNA by using a fluorescent probe-based TaqMan RT-PCR assay system. Components of the Fosun reagent include dNTPs, MgCl_2_, primers and probe, Taq DNA polymerase, reverse transcriptase, and the UNG enzyme. Primers and probe were designed to detect three target genes based on the original strain: E, N, and ORF1ab. The limit of detection (LOD) was 300 copies/mL with a Ct cut-off value < 36. The clinical evaluation of Fosun was conducted with 597 specimens and, compared with the comparator kit, the sensitivity was 99.51%, the specificity was 96.44%, the accuracy was 97.49%, the positive predictive value was 93.55%, and the negative predictive value was 99.74% [26]. The amplification time was 1 h and 20 min. This study aims to highlight the potential benefits of combining S2 with target genes in a commercial kit for detecting SARS-CoV-2 virus and the one with multiple mutations. 

## 2. Materials and Methods

### 2.1. Clinical Specimens and Nucleic Acid Extraction

A total of 694 leftover respiratory specimens from suspected patients during the outbreak (2019–2022) including B.1.1.7 (Alpha), B.1.351 (Beta), B.1.617.2 (Delta), B.1.1.529 (Omicron), omicron subvariants (BA.1, BA.2, BA.3 and BA.4), and Thailand’s local lineages were obtained from clinical specimens diagnosed at the Thai Red Cross Emerging Infectious Disease Health Science Centre (TRC-EID). Nucleic acid was extracted using magLEAD^®^ Consumable Kit (Precision System Science, Chiba, Japan) according to the manufacturer’s instructions. Isolated nucleic acid was used as a template for real-time RT-PCR and stored at −80 °C until further use. Nucleic acid of other human coronaviruses used in the validation of S2 specificity was checked for quality before testing. 

### 2.2. Primers and Probe Design

Primers and probe specific to S2 were designed to target conserved positions on the connector domain between HR1 and HR2 (985–1162) by using the Primer3 software (4.1.0) (Figure 1A) (https://primer3.ut.ee/ (accessed on 3 January 2024)). The S2 primers and probe were used in combination with three specific target genes (ORF1ab, N, E) in Fosun kit. The S2 probe was labelled with fluorescent dye, Cy3, to avoid the overlapping of emission spectra from other fluorescent dyes (FAM-ORF1ab, JOE-N, ROX-E, and CY5-IC) in the kit. The sequences were as follows: spike S2F; 5′-TCCCTCAGTCA GCACCTCAT-3′ and S2R; 5′-CTCGTGAAGGTGTCTTTGTTTC-3′ as primers, and S2Pr; 5′-Cy3-AACTGCTCCTGCCATTTGTC-3′ as the fluorescent probe.

### 2.3. The Validation of S2 Primers and Probe

One microliter of 10 μM spike S2F, S2R primer and S2Pr probe were added in combination with E, N, and ORF1ab gene in Fosun reagent. Real-time RT-PCR was performed according to the manufacturer’s instructions using the ABI 7500 Fast (Applied Biosystems, Foster City, CA, USA). LOD was evaluated twice using 10 μL of S2 control plasmid (ranging from 10^10^ to 10^0^ copies/mL). The validation of S2 specificity was confirmed by testing with other human coronaviruses including one coronavirus-229E (Ct = 20.9), three coronaviruses-OC43 (Ct = 16.0, 24.4, and 32.6), and two coronaviruses-NL63 (Ct = 21.2, 24.0). The evaluation of adding S2 in combination with target genes in the commercial kit was performed using nucleic acids from 694 suspected patients during the outbreak. 

### 2.4. Checking for Mismatch in Primers and Probe 

To find out about mutations on S2 primers and probe, 230,821 nucleotide sequences of SARS-CoV-2 submitted from January to October 2022 by 20 representative countries in each region from Global Initiative on Sharing All Influenza Data (GISAID) database were analyzed by using PrimerChecker (v3.04) (https://gisaid.org/ (accessed on 3 January 2024)). The listed representative counties include United States, Canada, Australia, Germany, France, United Kingdom, Italy, Russia, China, Japan, South Korea, India, Iceland, Brazil, Spain, Belgium, Turkey, Peru, Chile, and Thailand. In addition, representative nucleotide sequences of the top 5 currently circulating variants in all countries and Thailand from June to November 2023 including EG.5.1.1, EG.5.1, FL.1.5.1, FL.4, GE.1, HK.3, HV.1, JN.1, XBB.1.5, XBB.1.9.1, XBB.1.9.2, XBB.1.16, XBB.1.16.1, XBB.1.16.2, and XBB.2.3 were aligned with S2 primers and probe using MEGA-X. 

### 2.5. Statistical Analysis

One-way ANOVA with a significance level of 95% using SPSS was performed to determine whether addition of S2 primers and probe to Fosun kit affected the Ct value of E, N, and ORF1ab or not. Mean Ct value of each gene from the same clinical specimens were compared between Fosun real-time RT-PCR kit and the kit in combination with S2 primers and probe.

### 2.6. Ethical Statement

The study was conducted according to the guidelines of the Declaration of Helsinki and approved by the Institutional Review Board at the Faculty of Medicine, Chulalongkorn University (IRB number 400/63).

## 3. Results

### 3.1. The Validation of S2 Primers and Probe

To determine whether the newly designed S2 primers and probe were suitable for SARS-CoV-2 detection, real-time RT-PCR was performed twice using 10^10^ to 10^0^ copies/mL of SARS-CoV-2 S2 control plasmid. The LOD of real-time RT-PCR targeting S2 was 10^3^, which was equivalent to 333 copies/mL with a Ct cut-off value ≤ 34. Meanwhile, the LOD of E, N, and ORF1ab genes in the Fosun kit was 300 copies/mL with the Ct cut-off value ≤ 36, according to the instructions. The results show that S2 had slightly low sensitivity compared to E, N, and ORF1ab target genes (Appendix A) but there were no statistically significant differences between the Ct value of each gene; E, N, ORF1ab, and S2 (Figure 1B). A comparison of the mean Ct value of the same specimens tested with S2 primers and probe in combination with Fosun and Fosun alone showed no significant differences between the reagents (Figure 1C). Interestingly, delayed amplification in ORF1ab gene was detected in one specimen (omicron subvariant BA.4) from 694 specimens, whereas amplification problems were not encountered in the S2 target (Figure 1D). Furthermore, 77 of 694 specimens determined the nucleotide sequences by next-generation sequencing (NGS) did not find mutations in the S2 primers and probe. In addition, the real-time RT-PCR targeting of S2 did not cross-react with other human coronaviruses including human coronavirus-229E, -OC43, and -NL63 (Figure 1E).

### 3.2. Mutations within the Binding Site of S2 Primers and Probe

We analyzed mismatch pairing in primers and probe with 230,821 sequences of SARS-CoV-2 genome from the GISAID database submitted between January and October 2022 by 20 representative counties in each region around the world by using PrimerChecker (v3.04) (Appendix A). Mutations in the binding sites of primers and probe were found in various patterns, e.g., single-point mutation or multiple-point mutations, mutations in the 3′ end or mutations in other regions of primers and probe. Mutations in the S2F primer were found in 330 sequences, the S2R had 367 sequences, and the S2Pr probe had 1138 sequences of 230,821 sequences (Appendix A). The nucleotide sequences showing significant mutation that may affect the test are shown in Table 1. We found the most affected sequences in the S2F primer were BA.1.1.7 and unassigned lineages from India, the most affected sequences in S2R primer were unassigned lineages from the USA, and the most affected sequences in the S2Pr probe were BA.1 lineages from Turkey, respectively.

Multiple alignments of nucleotide sequences of the top five currently circulating lineages in all countries and Thailand from June to November 2023 showed that S2 primers and probe are perfect matches with XBB.1.5, XBB.1.9.1, XBB.1.9.2, XBB.1.16, XBB.1.16.1, XBB.1.16.2, XBB.2.3, EG.5.1, EG.5.1.1, FL.1.5.1, FL.4, GE.1, HK.3, HV.1, and JN.1 without mismatch recognition (Figure 2).

## 4. Discussion

Several commercial kits such as the TaqPath COVID-19 Combo Kit, FTD SARS-CoV-2 qPCR Test, QiaStat-Dx Respiratory SARS-CoV-2 Panel, Allplex 2019-nCoV, SARS-CoV-2 R-GENE, and Gene Xpert Xpress SARS-CoV-2, etc. have been found to have amplification problems in detecting current omicron variants [11,13,14]. Using the RT-PCR kit with a low limit LOD of ~100 copies/mL is useful in detecting viruses during the early stages of infection with low viral load. However, it may also delay hospital discharge or isolation where negative results are required, as lower-limit LOD kits often detect target genes in cured patients without clinical significance. Therefore, it is important to have proper LOD concentration to detect the infective stage of the virus. The commercial Fosun kit with 300 copies/mL LOD was authorized for emergency use by the U.S. FDA and has not reported any performance issues. We selected the Fosun kit for detecting SARS-CoV-2 in our laboratory [11]. The Fosun kit is suitable for our usage. The results from the Fosun kit were concordant with the symptoms of the patients. In addition to the proper LOD, the purpose of use, time, the number of samples, and expenses are considered helpful. In this experiment, we simultaneously detected E, N, ORF1ab, and S2 genes of SARS-CoV-2 with the Fosun reagent to decrease the resource burden on laboratory operations and save time. The LOD of S2 was 333 copies/mL with a Ct cut-off value < 34 while the LOD of E, N, and ORF1ab genes in the kit was 300 copies/mL with the Ct cut-off value ≤ 36. The slightly low sensitivity of S2 compared to E, N, and ORF1ab target genes may be because reagents and conditions were adapted to be equally suitable for amplifying E, N, and ORF1ab while S2 must use reagents and conditions according to the Fosun kit. Although S2 had a slightly low sensitivity, there were no significant differences between the mean Ct value of each target gene (Figure 1B), revealing that S2 is a promising target for detection as the same other target genes in the commercial kit.

Interestingly, one specimen (omicron subvariant BA.4) from 694 specimens showed a late amplification in ORF1ab in Fosun (Figure 1D). This is evidence that the addition of S2 as the target in combination with a commercial kit improves diagnostics even though some targets failed. The addition of S2 to the available commercial kit can be used to detect variants with extremely high mutations, such as omicron and subvariants, and it exhibited accuracy, reliability, sensitivity, and specificity the same as that of the kit. The comparison of the mean Ct value of the same specimens tested with S2 primers and probe in combination with Fosun and Fosun alone showed no significant differences between the reagents (Figure 1C). It indicated that the addition of S2 primers and probe does not affect the sensitivity of the reaction.

This is according to results from the PrimerChecker-analyzed mismatch pairing in primers and probe with 230,821 genome sequences. Diverse mutation patterns were found in the binding sites of primers and probe. Mutations in the S2F primer were found in 330 sequences (0.1430%), the S2R had 367 sequences (0.1590%), and the S2Pr probe had 1138 (0.4930%) sequences of 230,821 sequences (Appendix A). However, not all mutations will affect testing. The impact of mismatch binding on real-time RT-PCR depends on the number of point mutations and the distance from the 3′ end of primers and probe [27,28,29]. Significant mutations in nucleotide sequences submitted from January to October 2022 by 20 representative counties may possibly cause false negatives in S2 primers and probe in diagnostic tests, as shown in Table 1. The most affected sequences in the S2F primer are BA.1.1.7 and unassigned lineages from India. These sequences have five mutation points in the 3′ end of the S2F primer with 0.0004% of 230,821 submitted sequences. The most affected sequences in the S2R primer are unassigned lineage from USA, with four mutations in the 3′ end of the primer with 0.0004% of 230,821 sequences. The most affected sequence in S2Pr probe is the BA.1 lineage from Turkey, with four mutations in the 3′ end of the probe with 0.0004% of 230,821 sequences. Although we found affected sequences, the chance of mismatches reducing PCR performance is very low due to finding sequences with high mutations being very rare (0.0004%). 

N764K, D796Y, Q954H, N969K, T1100T, H1101D/Y, D1118H, G1124V, V1176F, Q1201H, V1228L, and M1237I mutations, respectively, were found in the S2 subunit [17,23,25,27,28]. However, these mutations did not impact S2 primers and probe because their positions were located on the connector domain between HR1 and HR2 (1053–1097) (Figure 1A). No essential mutations have been reported in this area [10,17,18,21,23,24,25]. Since the Public Health Ministry of Thailand has declared COVID-19 to be endemic and has approved the use of self-antigen test kits (ATK) without reconfirming via RT-PCR since July 2022, leading to a lack of specimens for the evaluation of real-time RT-PCR, we targeted S2 according to the variants circulating over time. In addition, there is always the possibility that certain mutations could increase in the future. The continued tracking of SARS-CoV-2 variants is necessary. To solve this issue, multiple nucleotide alignments of the top five currently circulating variants in all countries and Thailand (Jun–Nov 2023) with S2 primers and probe were performed to check for mismatch. The alignment results showed that S2 primers and probe are a perfect match with XBB.1.5, XBB.1.9.1, XBB.1.9.2, XBB.1.16, XBB.1.16.1, XBB.1.16.2, XBB.2.3, EG.5.1, EG.5.1.1, FL.1.5.1, FL.4, GE.1, HK.3, HV.1, and JN.1 without mismatch recognition (Figure 2). The results from PrimerChecker and sequence alignment were concordant with the results from real-time RT-PCR which have not encountered any amplification problems. 

## 5. Conclusions

The emergence of new SARS-CoV-2 variants with high numbees of mutations in several genes may lead to false-negative results in the COVID-19 real-time RT-PCR test. It is necessary to improve existing methods to solve this problem. The spike S2 subunit is markedly conserved and can be utilized for detecting variants with extremely high mutations, such as omicron and subvariants. The addition of S2 primers and probe to the available commercial kit exhibited accuracy, reliability, sensitivity, specificity, and ease of use. Additionally, recent vaccine developments using the conserved S2 subunit as a target elicits strong broad protection against SARS-CoV-2 variants and other coronaviruses and synergizes with B cells, leading to lifelong immunity [30,31]. Our study showed that the spike S2 could be a promising target for the detection of SARS-CoV-2 variants with extremely high numbers of mutations. Interested parties can apply S2 primers and probe to use according to the suitability of the lab by changing the reagents and validating the protocol.

## Figures and Tables

**Figure 1 tropicalmed-09-00050-f001:**
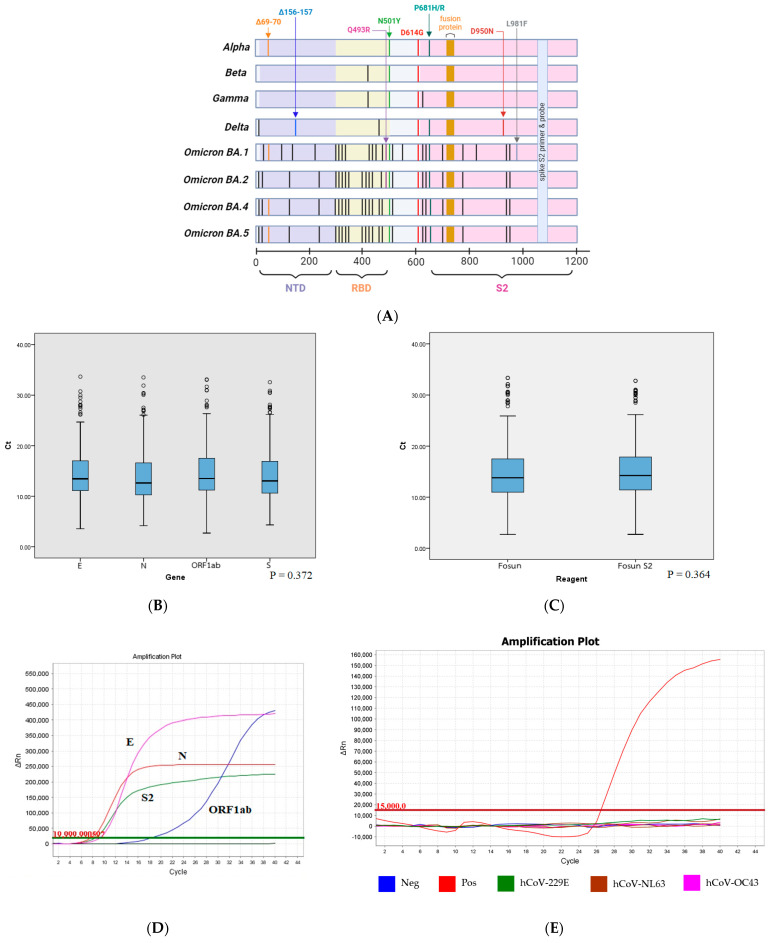
Real-time RT-PCR targeting S2 for detection of further SARS-CoV-2 variants with extremely high mutation rates. (**A**) SARS-CoV-2 spike protein structure and critical amino acid changes in VOCs and omicron subvariants. (**B**) Comparison of the mean Ct value of each target gene (E, N, ORF1ab, and S2) from 694 specimens; there were no significant differences between target genes (*p* > 0.05). (**C**) Comparison of the mean Ct vale of the same specimens (*n* = 694) tested with Fosun and Fosun S2; there were no significant differences between the reagents (*p* > 0.05). (**D**) Delayed amplification in ORF1ab gene in one BA.4 omicron. (**E**) Real-time RT-PCR targeting S2 did not cross-react with other human coronaviruses.

**Figure 2 tropicalmed-09-00050-f002:**
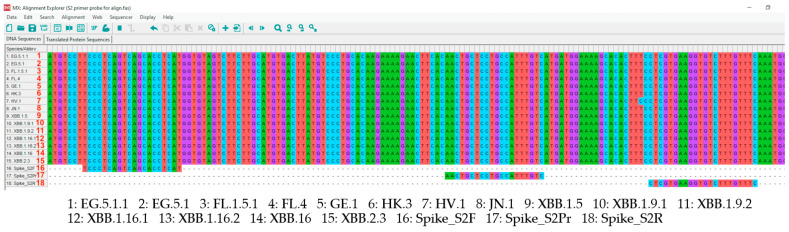
Multiple nucleotide alignment of top five currently circulating variants in all countries and Thailand (Jun–Nov 2023) with S2 primers and probe.

**Table 1 tropicalmed-09-00050-t001:** Significant mutations in nucleotide sequences may possibly cause false negatives in S2 primers and probe.

Primer	Lineage	Country	Mutation in Primer	Mutation in 3′ End	Number Sequences	Frequency (Total 230,821)
**S2F**	BA.1.1.7	India	18	5	1	0.0004%
	Unassigned	India	5	5	1	0.0004%
	B.1.1	Turkey	4	4	1	0.0004%
	B.1.1	Turkey	3	3	1	0.0004%
	BA.1	Turkey	3	2	1	0.0004%
	B.1.1	Turkey	2	2	1	0.0004%
	BA.5.3.2	Germany	4	1	1	0.0004%
			1	1	65	0.0282%
**S2R**	Unassigned	USA	4	4	1	0.0004%
	B.1.1.529	USA	4	3	1	0.0004%
	BA.2	Turkey	3	3	1	0.0004%
	BA.1	Turkey	3	1	1	0.0004%
			1	1	171	0.0741%
**S2Pr**	BA.1	Turkey	4	4	1	0.0004%
	BE.1	Turkey	3	3	1	0.0004%
	BA.2.12.1	USA	3	3	1	0.0004%
	BA.1	USA	2	2	1	0.0004%
			1	1	44	0.0191%

## Data Availability

Data is unavailable due to privacy or ethical restrictions.

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
