# Peer review of "Spike S2 Subunit: Possible Target for Detecting Novel SARS-CoV-2 Variants with Multiple Mutations"

_tropicalmed, 2024, doi:10.3390/tropicalmed9020050_

Round 1
Reviewer 1 Report
Comments and Suggestions for Authors
Dear Authors: This is very interesting and valuable work. The authors demonstrated that the S2 subunit may be a promising target for the detection of SARS-CoV-2 variants taking into account the markedly conserved S2 subunit. The SARS-CoV 2 variants have high rate mutations that impact the molecular diagnosis and it could be a good alternative to include these primers in association with other regions in the same reaction. Some questions can improve the understanding of the work as mentioned: 1-In the legend of Figure 1 it is not mentioned that the results came from 649 samples; Is this correct? 2-There is no mention of the FOSUN kit/reagents: What is the content? 3-Online 148-149. It is mentioned that the S2 subunit did not cross-react with other human coronaviruses. However, this result is not shown as mentioned in lines 109-110 (Mat and Met.).
Author Response
Comments 1: In the legend of Figure 1 it is not mentioned that the results came from 649 samples; Is this correct?
Response 1: Yes, the results in Figure 1 are based on an evaluation of 694 samples. We already referred to "694 specimen", (n=694) in the caption of Figure 1 on page3.
Comments 2: There is no mention of the FOSUN kit/reagents: What is the content?
Response 2: Thank you for pointing this out. I have already added the components of the Fosun reagent on page 2, paragraph 3, and in the introduction on lines 85-86.
Comments 3: Online 148-149. It is mentioned that the S2 subunit did not cross-react with other human coronaviruses. However, this result is not shown as mentioned in lines 109-110 (Mat and Met.).
Response 3: Thank you for your valuable comment. I have amended Figure 1 by presenting the results in Figure 1E and added content on nucleic acids of other human coronaviruses on page 4 lines 109-110 of the materials and methods part 2.1 Clinical specimens and nucleic acid extraction. Content about other human coronaviruses has been added to materials and methods part 2.3 Primers and probe validation on page 4 lines 126-128.

Reviewer 2 Report
Comments and Suggestions for Authors
Finding patients by testing is very important for both quarantine and treatment. PCR is the method of first choice for this purpose because of its high sensitivity and the fact that it does not produce false positives. However, a major concern is that primers will become unusable as viruses continue to mutate. This is because certain variants will no longer be detectable.
This paper shows that the addition of the S2 subunit to the conventional primers considerably alleviates this concern. This is an encouraging and important finding.
However, there is no guarantee that this can be used indefinitely in the future, and this should be taken into account in the discussion. As indicated here, there is always the possibility that certain mutations will increase false negatives. This will require constant attention.
And the fact that the S2 subunit was shown to be conserved here may be a useful suggestion for vaccine production. Vaccines targeting the spike have quickly lost their potency. Alternatively, a vaccine targeting S2 could extend the life of the product. It would be appreciated if this could also be mentioned in the discussion.
Author Response
Comment 1: There is no guarantee that this can be used indefinitely in the future, and this should be taken into account in the discussion. As indicated here, there is always the possibility that certain mutations will increase false negatives. This will require constant attention.
Response 1: Thank you for pointing this out. I completely agree with you, so I would like to add your comment to the discussion on page 7, lines 248-252.
Comment 2: The S2 subunit was shown to be conserved here may be a useful suggestion for vaccine production. Vaccines targeting the spike have quickly lost their potency. Alternatively, a vaccine targeting S2 could extend the life of the product. It would be appreciated if this could also be mentioned in the discussion.
Response 2: Thank you for your valuable feedback. I have added the fact that the S2 subunit was shown to be conserved here may have useful implications for vaccine production in the conclusion on page 8, lines 265-267.

Reviewer 3 Report
Comments and Suggestions for Authors
Manuscript ID tropicalmed-2822209 by Ponpinit T et al. entitled "Spike S2 Subunit: Possible Target for Detecting Novel SARS-CoV-2 Variants with High Mutations' is an original paper that requires numerous corrections.
L51.-Review this statement, bearing in mind that it may increase in some cases and not in others. "Genetic mutation in SARS-CoV-2 increases the severity of the disease and its ability to spread from human to human".
L74.- The authors should review and discuss recent publications on mutations in the S2 subunit of SARS-CoV-2, e.g. Kumar S, doi: 10.1128/jvi.00922-23, Abavisani M, doi: 10.1186/s12985-022-01951-7.
L79.- Given the method used by the authors (the limit of detection (LOD) was 300 copies/ml with a Ct cut-off < 36), they should discuss how their proposal is affected when the limit of detection (LoD) is greater than 100 copies of viral RNA (Arnaout, doi: 10.1101/2020.06.02.131144).
L81, 110, 128, etc. - Its wording is confusing. Did the authors add the whole S2 subunit or did they add S2 subunit primers? If it is the first (less likely), what were the characteristics of the S2 subunit that the authors added?
P3.- The information at the bottom of Figure 1 should be more precise.
The limitations of the research should be stated in the discussion section.
Author Response
Comments 1: L51.-Review this statement, bearing in mind that it may increase in some cases and not in others. "Genetic mutation in SARS-CoV-2 increases the severity of the disease and its ability to spread from human to human".
Response 1: Thank you for your valuable comment. I agree with this comment. Therefore, I have amended the content about genetic mutations on page 2, lines 45-49.
Comments 2: L74.- The authors should review and discuss recent publications on mutations in the S2 subunit of SARS-CoV-2, e.g. Kumar S, doi: 10.1128/jvi.00922-23, Abavisani M, doi: 10.1186/s12985-022-01951-7.
Response 2: Thank you for your valuable comment. I have reviewed and discussed the mutations in the S2 subunit following your suggestion in the introduction on page 2, lines 74-80, and the discussion on page 7 lines 240-242.
Comment 3: L79.- Given the method used by the authors (the limit of detection (LOD) was 300 copies/ml with a Ct cut-off < 36), they should discuss how their proposal is affected when the limit of detection (LoD) is greater than 100 copies of viral RNA (Arnaout, doi: 10.1101/2020.06.02.131144).
Response 3: Thank you for pointing this out. I have reviewed and discussed the LOD in discussion on page 6, lines 195-205.
Comment 4: L81, 110, 128, etc. - Its wording is confusing. Did the authors add the whole S2 subunit or did they add S2 subunit primers? If it is the first (less likely), what were the characteristics of the S2 subunit that the authors added?
Response 4: Thank you for pointing this out. I added S2 primers and probe to the Fosun reagent which was designed to target conserved positions on the connector domain between HR1 and HR2 in S2 subunit. All in the manuscript, I have amended it by changing S2 subunit to S2 primer and probe for correct understanding.
Comments 5: P3.- The information at the bottom of Figure 1 should be more precise.
Response 5: Thank you for pointing this out. I have amended the caption of Figure 1 on page 3.
Comments 6: The limitations of the research should be stated in the discussion section.
Response 6: Thank you for your valuable comment. I have discussed the limitation in the discussion on page 7, lines 244-255.

Reviewer 4 Report
Comments and Suggestions for Authors
The manuscript describes the evaluation of an RT-PCR kit for the detection of SARS-CoV-2 (including mutants) using “traditional” target genes (E, N, ORF1ab) enriched with a new target, the S2 subunit. The manuscript was sent in the form of a Brief Report, which in the reviewer's opinion is reasonable. The text is well written, the structure of the manuscript is consistent logically.
In the reviewer's opinion, the manuscript should be published with minor revisions:
1. Please shorten Introduction: the sentences in lines 36-41 are common knowledge and not directly related to the conducted research.
2. When entering the names of reagents, kits, the city of the company's headquarters should be added in parentheses in addition to the name of the manufacturer and the country.
3. Read the text, paying attention to editorial corrections, e.g.: line 39 - period before the link, line 72 - "(Figure 1A)" should be before the link, line 97 - correct the font size of the website.
Author Response
Comment 1: Please shorten Introduction: the sentences in lines 36-41 are common knowledge and not directly related to the conducted research.
Response 1: Thank you for pointing this out. I agree with this comment. Therefore, I have revised by shortening the introduction. The sentences in lines 36-41 were removed.
Comment 2: When entering the names of reagents, kits, the city of the company's headquarters should be added in parentheses in addition to the name of the manufacturer and the country.
Response 2: Thank you for your comment. I have added the city of the company's headquarters in addition to the name of the manufacturer and the country on lines 83, 107, and 124-125.
Comments 3: Read the text, paying attention to editorial corrections, e.g.: line 39 - period before the link, line 72 - "(Figure 1A)" should be before the link, line 97 - correct the font size of the website.
Response 3: Thank you for pointing this out. I mentioned "(Figure 1A)" in lines 70-71 because I want to show the parts of spike protein that I described in the introduction. In addition, I have added "(Figure 1A)" as you mentioned followed by the link on line 113 and lines 243-244 to emphasize the positions of primers and probe. I have changed the font size.

Round 2
Reviewer 3 Report
Comments and Suggestions for Authors
Out of a total of 694 respiratory samples, how many were found with mutations in S2? And how did the authors confirm these?
What are the sensitivity, specificity and predictive value of the Fosum kit?
Author Response
Comment 1: Out of a total of 694 respiratory samples, how many were found with mutations in S2? And how did the authors confirm these?
Response 1: Thank you for pointing this out. I have added “Furthermore, 77 of 694 specimens determined the nucleotide sequences by next-generation sequencing (NGS) did not find mutations in S2 primers and probe” in the results part on page 5, lines 172-174. In addition, I would like to add “Mutations in S2F primer were found in 330 sequences, the S2R had 367 sequences, and the S2Pr probe had 1,138 sequences of 230,821 sequences (Table S2)” in the results part on page 5, lines 182-184 and “Mutations in the S2F primer were found in 330 sequences (0.1430%), the S2R had 367 sequences (0.1590%), and the S2Pr probe had 1,138 (0.4930%) sequences of 230,821 sequences (Table S2). However, not all mutations will affect testing” in the discussion part on page 7, lines 235-238. I also modified table 1, table S1, and table S2 to summarize all data.
Comment 2: What are the sensitivity, specificity and predictive value of the Fosum kit?
Response 2: Thank you for pointing this out. I completely agree with you about giving the data about sensitivity, specificity, and predictive value of the Fosum kit, so I would like to add “Clinical evaluation of Fosun was conducted with 597 specimens and compared with the comparator kit, the sensitivity was 99.51%, the specificity was 96.44%, the accuracy was 97.49%, the positive predictive value was 93.55%, and the negative predictive value was 99.74% [26]” in the introduction part on page 2, lines 89-92.
